# OpenReview forum: "Operationalizing a Threat Model for Red-Teaming Large Language Models (LLMs)"
_TMLR — Accepted by TMLR_

### Review · Reviewer_6QJR · 2025-01-05

**Summary Of Contributions:**

This paper proposes a systematical survey to existing red-teaming attacks on LLMs. The covered topic is exhaustive and the provided comments and analysis are sound.

I believe this survey provides a timely and in-depth analysis towards LLM safety and should be of great value to the community.

**Audience:**

Yes

**Claims And Evidence:**

Yes

**Requested Changes:**

See above.

**Strengths And Weaknesses:**

**Stengths:**

* All the red-teaming attacks (as far as I konw) for LLMs are covered in this paper. It requires a lot of hard work and also need a very large volumn of reading.

* The authors provide their own insights and in-depth analysis to the attacks, which can not be achieved without a good understanding of the field.

* Papers are well-structured and easy to read.  The introduction is extremely well-written and it is an interesting read. Section 2 provides a ccomprehensive backgound of LLM, definition of harmful behaviors and motivate the need of red-teaming evaluation. I like the authors' viewpoints of taking red-teaming practice as a form of LLM evaluation.

* Figure 2 is really useful and comprehensive. This simple figure covers all the red-teaming practices comprehensively as far as I know.

**Weakness:**
* Section 3.2, it is recommended to separately list out the adversary capability for each  red-teaming practice listed in Section 3.1.

* Section 3.3, it is also recommended to separately list out the goals of each attack vector.

* Section 4.2.1, I think this section should be merged with Section 4.1, Jailbreak attack. It seems that Perez et al. and Zou et al are standard  jailbreak technique. A suggestion is that maybe you can make a better catagory under jailbreak attack 4.1, and merge those discussion on 4.2.1 into this section. I think the main the problem probably is that Section 4.1 jail-break attack actually has a great overlap with direct attack, or potentially is one the sub-category of the defined direct attack. The authors should find a way to avoid this confusion in terms of category issue.

* Section 4.3, how infusion attack differs from jailbreak attack? I personally think they are the same because both of them manipulate the input of the data to bypass alignment. In context exampls do not differ much from the the normal prompt, and therefore I think it should be classified as one of the jailbreak attack as well.

* Section 4.4, it seems that jailbreak attack is also one of the inference-time attack because it also takes place in inference-time. I know your definition of inference time attack is that the attacker have control over the the model weights and thus the activation. Perhaps it should be instead call  while box inference -time attack and the previous version is called blackbox inference-time attack?

* Section 4.5.3, why GCG and AutoDan is classified as training time attack again? It does not make sense at all.  Moreover, I don't understand what do you refer to gradient-based attack in this section. I think this section should be removed.

* Section 4.5.2, recently there is a few literature on backdoor attack/defense for LLM. They should be included and discussed

Probe before You Talk: Towards Black-box Defense against Backdoor Unalignment for Large Language Models (ICLR annonynmous submission)

BackdoorLLM: A Comprehensive Benchmark for Backdoor Attacks on Large Language Models

* Section 4.5.1, the title should be changed to "harmful fine-tuning attack" instead of "alignment erasure".

* Section 4.5.2, since (Qi et al,2023),  there are recently a surge of research paper on harmful fine-tuning attacks, as follows. I encourge the authors to discuss these recent papers.

Shadow Alignment: The Ease of Subverting Safely-Aligned Language Models arXiv

Fine-tuning aligned language models compromises safety, even when users do not intend to! ICLR 2024

On the Vulnerability of Safety Alignment in Open-Access LLMs ACL2024 (Findings)

Lora fine-tuning efficiently undoes safety training in llama 2-chat 70b SeT LLM workshop@ ICLR 2024

Removing RLHF Protections in GPT-4 via Fine-Tuning NAACL2024

What's in your" safe" data?: Identifying benign data that breaks safety COLM2024

Covert malicious finetuning: Challenges in safeguarding llm adaptation ICML2024

The effect of fine-tuning on language model toxicity NeurIPS2024 Safe GenAI workshop

Towards Understanding the Fragility of Multilingual LLMs against Fine-Tuning Attacks arXiv

PEFT-as-an-Attack! Jailbreaking Language Models during Federated Parameter-Efficient Fine-Tuning arXiv

Harmful Fine-tuning Attacks and Defenses for Large Language Models: A Survey arXiv

Emerging Safety Attack and Defense in Federated Instruction Tuning of Large Language Models



* Section 5.2, recently there are a few defenses towards harmful fine-tuning attack. Please discuss the following relevant paper and also update Table 2 to include these relevant works.

Vaccine: Perturbation-aware alignment for large language model aginst harmful fine-tuning NeurIPS2024

Representation noising effectively prevents harmful fine-tuning on LLMs NeurIPS2024

Buckle Up: Robustifying LLMs at Every Customization Stage via Data Curation ICLR2025 Submission

Tamper-Resistant Safeguards for Open-Weight LLMs ICLR2025 Submission

Booster: Tackling harmful fine-tuning for large language models via attenuating harmful perturbation ICLR2025 Submission

Leveraging Catastrophic Forgetting to Develop Safe Diffusion Models against Malicious Finetuning NeurIPS2024

 Identifying and Tuning Safety Neurons in Large Language Models ICLR2025 Submission

Targeted Vaccine: Safety Alignment for Large Language Models against Harmful Fine-Tuning via Layer-wise Perturbation arXiv

Fine-tuning can cripple your foundation model; preserving features may be the solution TMLR

Safety-Tuned LLaMAs: Lessons From Improving the Safety of Large Language Models that Follow Instructions ICLR2024

Safety fine-tuning at (almost) no cost: A baseline for vision large language models ICML2024

Assessing the brittleness of safety alignment via pruning and low-rank modifications ME-FoMo@ICLR2024

Mitigating fine-tuning jailbreak attack with backdoor enhanced alignment NeurIPS2024

Keeping llms aligned after fine-tuning: The crucial role of prompt templates NeurIPS2024

Lazy safety alignment for large language models against harmful fine-tuning NeurIPS2024

Safety alignment should be made more than just a few tokens deep ICLR2025 Submission

 Do as I do (Safely): Mitigating Task-Specific Fine-tuning Risks in Large Language Models ICLR2025 Submission

Bi-Factorial Preference Optimization: Balancing Safety-Helpfulness in Language Models ICLR2025 Submission

Safety Layers in Aligned Large Language Models: The Key to LLM Security ICLR2025 Submission

SEAL: Safety-enhanced Aligned LLM Fine-tuning via Bilevel Data Selection ICLR2025 Submission

Safety Alignment Shouldn't Be Complicated ICLR2025 Submission

SaLoRA: Safety-Alignment Preserved Low-Rank Adaptation ICLR2025 Submission

Towards Secure Tuning: Mitigating Security Risks Arising from Benign Instruction Fine-Tuning ICLR2025 Submission

Safety-Aware Fine-Tuning of Large Language Models NeurIPS 2024 Workshop on Safe Generative AI

RobustFT: Robust Supervised Fine-tuning for Large Language Models under Noisy Response arXiv

Defending Against Unforeseen Failure Modes with Latent Adversarial Training arXiv

A safety realignment framework via subspace-oriented model fusion for large language models KBS

MoGU: A Framework for Enhancing Safety of Open-Sourced LLMs While Preserving Their Usability NeurIPS2024

Safe lora: the silver lining of reducing safety risks when fine-tuning large language models NeurIPS2024

Antidote: Post-fine-tuning safety alignment for large language models against harmful fine-tuning arXiv

Locking Down the Finetuned LLMs Safety ICLR2025 Submission

Unraveling and Mitigating Safety Alignment Degradation of Vision-Language Models ICLR2025 Submission

Separate the Wheat from the Chaff: A Post-Hoc Approach to Safety Re-Alignment for Fine-Tuned Language Models arXiv

NLSR: Neuron-Level Safety Realignment of Large Language Models Against Harmful Fine-Tuning AAAI2025

Navigating the safety landscape: Measuring risks in finetuning large language models NeurIPS2024



*  Section 5,2, in addition to LAT and BEEAR,  Vaccine[1] and T-Vaccine[2] should also be classified as a similar technique under adversarial training category.  The authors should dicuss these two relevent works.

[1] Vaccine: Perturbation-aware alignment for large language model aginst harmful fine-tuning NeurIPS2024

[2] Targeted Vaccine: Safety Alignment for Large Language Models against Harmful Fine-Tuning via Layer-wise Perturbation arXiv

---

> ### Author Response · Authors · 2025-01-30
> **Response Reviewer 1**
>
> # Response to Reviewer Comments
>
> We thank the reviewer for their thoughtful feedback. Here are our responses to the main points:
>
> ## On Separating Sections 4.1 and 4.2.1
> While we understand these sections may appear to overlap, their separation reflects fundamental differences in attack requirements and sophistication. Jailbreak attacks need only application-level access and can be executed by non-technical users through manual prompt crafting. In contrast, Direct attacks require API access, programming expertise, and systematic approaches like gradient optimization. These distinctions matter for practitioners implementing defenses.
>
> ## On Infusion vs Jailbreak Attacks
> Though both ultimately interact with LLMs through prompts, infusion attacks target a fundamentally different vector - the retrieval/reference data that augments the model's knowledge. Unlike jailbreak attacks which are ephemeral, infusion attacks can persist across multiple queries once poisoned data enters the retrieval corpus. This distinction highlights unique security challenges in retrieval-augmented architectures.
>
> ## On Taxonomy Structure
> We've clarified our terminology by renaming several categories and maintaining organization by access level rather than attack timing. The taxonomy aims to clearly delineate attacks based on required capabilities, providing practitioners with an actionable framework for defense.
>
> ## Additional Updates
> We've added detailed tables mapping adversary capabilities and goals to attack vectors, incorporated new papers on backdoor attacks and fine-tuning defenses, and renamed sections for clarity. We plan to integrate the many valuable recent papers suggested by the reviewer in the camera-ready version.
>
> For a more detailed response, including specific paper citations and technical explanations, please see: https://docs.google.com/document/d/1ZoiEE_BVHYQD4qRnpYvGIhs1oO6LiYmG1kI_QEGARgs/edit?tab=t.0

---

> ### Comment · Reviewer_6QJR · 2025-02-02
> **Thanks for the rebuttal**
>
> Thanks for the rebuttal. While most of my concerns are successfully addressed, I still recognize an issue when reading your revision.
>
> For while box prompt search attack, I respectively disagree to put it under training-time attack. This attack has no relevance with model training at all. It should belongs to one subsection under jailbreak attack.
>
> In addition, I suggest the authors to update the paper directly in openreview. While I found that the external link for version comparison somehow useful, it lose some function compared to a pdf (e.g., hyperlink to a reference). Moreover, providing too many external link (and external material) is actually not good for rebuttal... Just some minor suggestions here.

---

> > ### Author Response · Authors · 2025-02-06
> >
> > Dear Reviewer,
> >
> > Thank you for your thoughtful feedback and for helping us improve the paper. We have spent considerable time contemplating and discussing your points, particularly regarding classifying white-box prompt search attacks. We appreciate you raising this concern, as it might have also been a source of confusion for other readers.
> >
> > To address this and make our categorization rationale crystal clear, we have added several clarifying paragraphs throughout the paper:
> >
> > 1. At the beginning of the Attacks section (Section 4), we now explicitly state:
> >    > "Our taxonomy organizes attacks based on the highest level of access required during either the planning or execution phases of an attack. Analogously, in the White-Box Prompt Search Attack (refer to Section § 4.5.2), the planning phase requires executing a costly adversarial prompt search algorithm, whereas the execution phase entails directly prompting the LLM with the identified adversarial prompt."
> >
> > 2. At the beginning of Training Attacks (Section 4.5), we clarify:
> >    > "While some Frontdoor Attack categories like 'Harmful Fine-tuning Attack' and 'Adapter & Model Weights' modify model parameters during training, others like 'White-Box Prompt Search Attack' leverage access to model weights to learn adversarial attack prompts through gradient-based optimization. Though their attack vector is a prompt, these prompts are learned through an adversarial optimization process that requires full model access during the planning phase, distinguishing them from black-box 'Jailbreak Attack' approaches in Section § 4.1 that only require application input level access during both planning and execution phases."
> >
> > 3. In the White-Box Prompt Attacks section (Section 4.5.2), we provide detailed explanation of why these attacks are categorized here, including their relationship to transferable attacks and the crucial distinction of requiring direct model access during the planning phase.
> > > "These attacks optimize adversarial prompts by backpropagating through the target model to maximize the harmful output score as measured by a safety classifier. Unlike Transferable Attacks (Section § 4.2.1) which optimize against proxy models, these attacks require direct access to the target model’s weights and architecture. Note that methods like GCG (Zou et al., 2023) and AutoDAN (Zhu et al., 2023) (discussed in Section § 4.2.1) could also be classified here if used directly on the main model rather than a proxy model. In Section § 4.2.1, we categorize methods that explicitly show transferability using comprehensive experimentation. In contrast, this section classifies other white-box optimization methods without such transferability experimentation. Nonetheless, fundamentally, both methods are quite similar as they search for adversarial prompts through a resource-intensive optimization procedure. Representative attacks here include Wichers et al. (2024) which uses the Gumbel-Softmax trick for backpropagation through the discrete sampling step of LMs (Jang et al., 2016; Maddison et al., 2016), while Perez et al. (2022); Deng et al. (2022) employs reinforcement learning to discover these harmful attack prompts. Alternative methods include coordinate ascent (Jones et al., 2023) and constrained decoding using Langevin dynamics (Xingang et al., 2024), both of which produce human-readable adversarial prompts, contrasting with the unnatural prompts discovered by other automated techniques. Once these attack prompts are identified during the planning phase, an adversary can attack the LLM system by employing these prompts in the execution phase via simple prompting. Since the planning phase requires white-box access to the model weights, we categorize these attacks as training attacks. In addition, the attacker might publicly distribute these exploit prompts, encouraging other malicious actors to incorporate them into their attacks."
> >
> > Additionally, we have removed all references to "training-time" in the paper and replaced them with "training-based" to avoid potential confusion. Our categorization is fundamentally based on the level of model access required by potential adversaries in different phases of attack development, rather than when the attack occurs in the model lifecycle.
> >
> > We thank you for holding us accountable to perfection and hope that these clarifications address your concerns. We believe these changes will help readers better understand our taxonomy's organizing principle and the rationale behind our categorization decisions.
> >
> > Regarding your suggestion about updating the paper directly in OpenReview - we appreciate this feedback and will ensure future revisions are uploaded directly to the platform. We have also updated the pdf diff with the new version to facilitate ease of reviewing.

---

> ### Comment · Reviewer_6QJR · 2025-02-06
> **More discussion**
>
> Hi authors,
>
> I am still not convinced of and could not understand the reason of classifying a few jailbreak attack methods under training attack. Yes, the methods you mentioned,e.g., GCG requires gradient descent on the input samples. However, it does not make much sense to classify them under the training-based attack category, instead of jailbreak attack, This is really confusing because when people talk about GCG, they will think about jailbreak immediately. instead of the training-based attack that you categorize it into.
>
> Yes, these methods may somehow fall under two categories under your definition. However, the authors should avoid such a overlapping and confusing classification with a better category method.
>
> I welcome more discussion, and our argument on this point will not affect my overall  rating. Feel free to disagree my point.

---

> > ### Author Response · Authors · 2025-02-06
> > **Response to More Discussion**
> >
> > We appreciate the reviewer's thoughtful comments and welcome this discussion. We acknowledge that the term "Jailbreak" has been used loosely in the literature, which may contribute to some confusion.
> >
> > Our categorization of the first attack type as "Jailbreak Attack" stems from its specific focus on attacks against LLM-based applications. In these scenarios, user input is encapsulated within a meta-prompt (or application prompt) that typically includes safety instructions or guidelines. The objective of these attacks is to craft user prompts that escape the confines of these meta-prompts - hence the term "jailbreak," drawing parallels to bypassing restrictions on mobile devices.
> >
> > This usage aligns with how some industry experts employ the term. For instance, Microsoft's AI Red-Teaming team uses similar terminology (see https://youtu.be/UKj5jj6fD_c?t=580).
> >
> > We acknowledge, however, that the term "jailbreak" has been colloquially applied to a wide range of prompt-based attacks in the literature, potentially leading to confusion. To mitigate this, we are willing to revise the "Jailbreak Attack" category name in the camera-ready version. We suggest **"Application Attack"** as a possible alternative, but we are open to and would appreciate the reviewer's recommendations for a more precise and fitting term.
> >
> > Also note that based on previous response, we moved GCG and AutoDAN under Transferable Attacks under Direct Attacks and not under Jailbreak Attacks. To learn these adversarial prompts the attacker would take a proxy model run this expensive prompt optimization and then attack some black-box LLM (e.g. through OpenAI API) with these attack prompts. Both of these papers discuss transferability of their attacks.
> >
> > - [GCG](https://www.semanticscholar.org/reader/47030369e97cc44d4b2e3cf1be85da0fd134904a)
> > - [AutoDAN](https://www.semanticscholar.org/reader/1227c2fcb8437441b7d72a29a4bc9eef1f5275d2)
> >
> > The papers which do not discuss Transferability have been placed under White-box Prompt Search Attack such as
> > - [Gradient-Based Language Model Red Teaming](https://www.semanticscholar.org/reader/409e0616a0fc02dd0ee8d5ae061944a98e9bd5a9)
> > - [Red Teaming Language Models with Language Models](https://aclanthology.org/2022.emnlp-main.225/)
> >
> > We believe that categorizing all White-Box Prompt Search Attacks under Transferable Attacks or Jailbreak Attacks would be inaccurate for two reasons. First, many of these papers do not explicitly claim transferability. Second, these attacks require white-box access to the model, which is inconsistent with our definitions of Jailbreak and Direct Attacks. These attacks fundamentally differ in their access requirements and methodologies.
> >
> > However, we recognize the potential for confusion in our current categorization. To address this, we are open to renaming the "Jailbreak Attack" category to "Application Attack" if the reviewer believes this would provide greater clarity and resolve any discomfort with the current terminology. We are committed to ensuring our taxonomy is as clear and accurate as possible.
> >
> > We sincerely appreciate the reviewer's patience, time, and thoughtful engagement during this rebuttal phase. Your detailed feedback and willingness to discuss these nuances are invaluable in refining our work. We are grateful for the opportunity to clarify our perspective and potentially improve our taxonomy based on this constructive dialogue.

---

> ### Comment · Reviewer_6QJR · 2025-02-07
> **Several concerns**
>
> Hi authors,
>
> I found the classification model still confusing. Yes, I understand that you want to make the 5 categories  based on accesss capacity of the LLM. But still there should be a significant revision of the category,   Let's look at Figure 1  for easy communication.
>
> 1. For first category jailbreak attack, I think it should be deleted. These papers mentioned should be moved to direct attack. I suggest not to separate application input and API. They are the pretty similar in my view. I understand your intention because API give you more permission, but I don't think this minor difference is something very important.  The existence of this category will confuse many readers.
>
> 2. For direct attack-> automated attacks, why it leads to a category without name and a category transferable attacks?
> 3. I strongly disagree that jailbreak attack refer only to attack in which attacker has only application level permission, no matter how. Based on your model, I suggest separate jailbreak attack into two categories, i.e., blackbox jailbreak and whitebox jailbreak. For whitebox jailbreak, i agree to put it under training attack.
>
> 4. What I want to convey is that: **let's be simple and don't create too many terms** (e.g., prompt search attack, transfer attack, automatated attack). This really confuse readers if they don't have knowledge of what is going on. These attack only has minor difference in my understanding.

---

> > ### Author Response · Authors · 2025-02-08
> > **# Response to Several Concerns**
> >
> > We deeply appreciate the reviewer's detailed feedback which has helped us significantly improve our taxonomy's organization and clarity. Below we address each concern and detail our revisions:
> >
> > ## Reorganization of Attack Categories
> >
> > We like your suggestion on Black-Box and White-Box Jailbreak Attacks, we renamed Jailbreak A. and moved it under Direct A.:
> >
> > ### 1. Direct Attacks
> > - **Black-Box Jailbreak Attacks**
> >   - Manual Attacks (formerly standalone Jailbreak Attacks)
> >   - Automated Attacks
> >     - Transferable Attacks
> > - Inversion Attacks (same as before)
> > - Side-Channel Attacks (same)
> >
> > ### 2. Training Attacks
> > - **White-Box Jailbreak Attacks** (as suggested, moved under Training Attacks)
> > - Other categories remain unchanged
> >
> > ## Importance of Manual vs. Automated Attack Distinction
> >
> > We maintain this distinction for practical red-teaming implication:
> >
> > - **Practical Red-Teaming Implications**
> >    - Manual Attacks:
> >      - Can be evaluated by domain experts (journalists, legal scholars, ethicists)
> >      - Don't require technical expertise
> >      - Bring valuable domain-specific insights
> >    - Automated Attacks:
> >      - Require technical practitioners
> >      - Need programmatic API access
> >      - Enable systematic probing of system boundaries
> >
> > ## Rationale for Our Taxonomy
> > As discussed in our Related Works section, prior taxonomies have organized attacks based on either the type of risk posed (which is particularly useful for policymakers evaluating model safety across different harm categories) or the methodology used in constructing the attack (which benefits researchers studying attack techniques).
> > Our taxonomy takes a different approach by focusing on attack entry points, with the specific goal of supporting effective red-teaming in practice. Some key advantages of this organization include:
> >
> > It helps practitioners map required expertise to different attack vectors, facilitating recruitment of appropriate red team members
> > It enables organizations to structure comprehensive red-teaming programs by systematically covering different access levels
> > It naturally guides the allocation of defensive resources by highlighting which entry points require what types of protection
> > It provides clear guidance on what skills and backgrounds are needed when building red teams - from domain experts for application-level testing to technical specialists for API-level probing
> >
> > This organization is particularly valuable for ML practitioners planning and executing red-teaming exercises. While existing taxonomies excel at categorizing risks or methodologies, our entry point-based approach directly supports the practical implementation of red-teaming programs.
> > We have worked to make this practical focus clearer throughout the paper, as it appears the reviewer may have been evaluating our taxonomy through the lens of previous organizations rather than its intended use case in supporting effective red-teaming exercises.
> >
> >
> > ## On Terminology
> >
> > While we understand the concern about too many terms, distinctions like Automated vs. Manual and Transferable Attacks capture important properties that guide practical implementation:
> >
> > 1. **Manual vs. Automated**: Different expertise requirements, testing approaches, and recruitment needs
> > 2. **Transferable Attacks**: Special significance as they can affect multiple models with minimal adaptation
> >
> > We've clarified these distinctions while maintaining focus on practical utility for red-teaming exercises.
> >
> > **Why earlier Jailbreak Attacks (now Manual Attacks) cannot be merged with Automated Attacks**
> > While the technical distinction between Application Input and API access may appear subtle from a computer science perspective, it has significant practical implications for organizing red-teaming exercises. Someone with application-level access can only interact through the user interface, making these attacks accessible to domain experts like journalists or ethicists who bring valuable perspectives but may lack programming expertise. In contrast, API-level access enables programmatic probing of model behavior, requiring technical practitioners who can write code and systematically explore vulnerabilities. This natural division helps organizations recruit diverse red-team participants and structure comprehensive security assessments. We have enhanced the paper's focus on these practical red-teaming considerations based on feedback from other reviewers.
> >
> >
> > ## Conclusion
> >
> > Your feedback has been invaluable in improving the paper's organization and clarity. We've worked hard to:
> > - Cover diverse angles and simplify the overall structure
> > - Maintain important practical distinctions
> > - Enhance usefulness for practitioners
> > - Added many citations and connected Attacks and Defenses better to Red-Teaming.
> >
> > Given these substantial improvements addressing your concerns, we respectfully request that you consider raising your rating for the paper.
> >
> > Thank you again for your thoughtful engagement that has helped strengthen the paper.

---

> > > ### Author Response · Authors · 2025-02-08
> > > **# Paragraphs Rewritten/Added Towards Resolving the Concerns**
> > >
> > > ### Direct Attacks
> > > ```
> > > Notably, many sophisticated Automated Attacks trace their lineage to simpler Manual Attacks discovered by
> > > the broader AI safety community. For instance, the “Do Anything Now” (DAN) jailbreak (Shen et al., 2023d),
> > > originally shared on Reddit by an AI safety enthusiast, inspired more systematic approaches like AutoDAN
> > > (Zhu et al., 2023) and AutoDAN-GA (Liu et al., 2023b). Similarly, manual refusal suppression techniques
> > > pioneered by early attackers laid the groundwork for automated methods like Greedy Coordinate Gradient
> > > (GCG (Zou et al., 2023)). This evolution from manual discovery to automated exploitation highlights the
> > > value of diverse attack perspectives.
> > > ```
> > >
> > > ```
> > > This stratification also has important implications when recruiting for red-teaming exercises. Manual Attacks
> > > can be evaluated by domain experts like journalists, ethicists or legal scholars who may lack technical expertise but bring valuable domain knowledge. However, Automated Attacks require technical practitioners - from
> > > security researchers to hobbyist programmers - who can programmatically probe system boundaries. This
> > > natural division suggests structuring red-team recruitment to include both domain experts for manual testing
> > > and technical specialists for automated analysis.
> > > ```
> > >
> > > ### Transferable Attacks
> > > ```
> > > A particularly potent subset of Automated Attacks are Transferable Attacks, which
> > > demonstrate consistent effectiveness across different model architectures and deployments. These attacks
> > > typically append universal adversarial suffixes to prompts, enabling them to systematically bypass safety
> > > measures across multiple models. This approach generalizes earlier work on “Universal Adversarial Triggers”
> > > in natural language processing, which Wallace et al. (2019) characterized as “input-agnostic sequences of
> > > tokens that trigger a model to produce a specific prediction when concatenated to any input from a dataset.”
> > > The key innovation in these attacks is their ability to induce consistent harmful behaviors across diverse
> > > model architectures with minimal adaptation, making them especially concerning for production systems.
> > > ```
> > >
> > > ```
> > > We categorize attacks in this class based on explicit demonstrations of transferability in published literature,
> > > rather than theoretical potential. While other Automated Attacks may well transfer effectively across
> > > models, we restrict this category to work that has rigorously validated such claims through comprehensive
> > > experimentation. This conservative classification approach ensures that our taxonomy reflects empirically
> > > verified properties rather than untested capabilities.
> > > ```
> > > ```
> > > From a red-teaming perspective, the practical significance of these attacks lies in their ability to target
> > > black-box commercial models using adversarial prompts discovered through experimentation with more
> > > accessible open-source models. Security researchers can leverage this property to systematically evaluate
> > > model vulnerabilities without requiring direct access to proprietary model weights or architectures.
> > > ```

---

> ### Comment · Reviewer_6QJR · 2025-02-08
> **The structure looks better!**
>
> Hi authors,
> Thanks for the revision. The structure looks better. I have a  few more comments: Let's look at figure 1 again,
>
> 1. What are the items in blackbox-jailbreak->automated attack->Wolf In Sheep’s Clothing, etc. You should give them a name and clarify how it differs from transferable attack. If there are no significnat difference, merge them with transferable attack.
>
> I am done with the jailbreak attack. Let's look at the training-based attack. It has some classification issue as well.
>
> 2. I disagree to put harmful fine-tuning attack under front door attack. The attack model in [1][2] allow users to upload harmful data to the API to launch harmful fine-tuning attack. It does not  require access to model weights.
>
> 3. I disagree the name of frontdoor attack/backdoor attack. Data-Centric attack and model-centric attack looks better.
>
> [1] Qi, Xiangyu, et al. "Fine-tuning aligned language models compromises safety, even when users do not intend to!." arXiv preprint arXiv:2310.03693 (2023).
> [2] Huang, Tiansheng, et al. "Harmful fine-tuning attacks and defenses for large language models: A survey." arXiv preprint arXiv:2409.18169 (2024).
>
> PS: Not to worry about the rating. I will give acceptance anyway. What I want to do here is to improve this paper such that it is valuable to the community, and will not mislead direction.

---

> > ### Author Response · Authors · 2025-02-10
> > **Thank you for the positive feedback**
> >
> > We thank the reviewer for their thoughtful feedback and suggestions to improve the paper. We have made the following changes in response:
> >
> > 1. We have moved the harmful fine-tuning attack under the Data-Centric Attack category. We agree that even though some papers experiment on open-source models, the primary attack vector is the fine-tuning data, making it more appropriate to classify under Data-Centric Attacks. We appreciate the reviewer pointing this out and sharing the survey paper by Huang et al., which we have now cited in our paper.
> >
> > 2. We have renamed "Backdoor" and "Frontdoor" Attacks to "Data-Centric" and "Model-Centric" Attacks respectively, as suggested.
> >
> > 3. We have renamed "Wolf in Sheep's Clothing" to the specific technique name "ReNeLLM" and have checked other entries to ensure they use specific technique names where applicable.
> >
> > 4. Regarding the distinction between Automated Jailbreak Attacks and Transferable Attacks, we agree that it makes sense to keep these categories separate due to their different characteristics. To clarify this distinction, we have added the following paragraph to the paper:
> >
> > >Automated Jailbreak Attacks and Universal & Transferable Attacks (discussed next) differ significantly in their planning and execution phases. Automated attacks operate through black-box API access, iteratively generating and testing prompts in real-time, prioritizing accessibility and rapid deployment. Conversely, Universal & Transferable Attacks separate planning and execution. The planning phase involves computationally expensive searches for adversarial suffixes using proxy models. Execution then deploys these pre-computed attacks via API calls. This separation is necessary because, despite high attack transferability, not all attacks discovered on proxy models will successfully transfer to the target model. The methodological difference is evident in practical requirements: Automated attacks can be executed from any machine with API access, while Transferable attacks require GPUs and resource-intensive algorithms like Greedy Coordinate Gradient (GCG). This disparity represents a trade-off between accessibility and potential exploit robustness, with automated methods favoring broader applicability at the potential cost of attack potency."
> >
> > We believe these changes address the reviewer's concerns and improve the clarity and accuracy of our taxonomy. We appreciate the reviewer's commitment to ensuring the paper provides valuable and accurate information to the community. We thank the reviewer for their patience and guidance throughout the review process in improving the paper. We are committed to making this work as valuable as possible to the research community. If there are any other aspects of the paper that could be further improved, please let us know.

---

> ### Comment · Reviewer_6QJR · 2025-02-11
> **More feedback**
>
> Hi authors,
>
> Thanks for the revision. I think you misunderstand one of my comments. Let's look at Figure 1.
> What I am confused is that why under Automated Jailbreak Attack, you further classifies the into two sub-categories, one with no name and one with Universal &Transferable Attacks. What is that category with no name and how they differs from Universal &Transferable Attacks?  From your  last comments, it seems that Universal &Transferable Attacks is not a sub-category of Automated Jailbreak Attack and they are parallelly different?  However, I think its parent items, i.e., automated attack and manual attack are just fine and there should not be a third category. This confusion should be fixed by some ways.
>
>
> Also, why under Inversion Attacks->Model Inversion, it is classifies as two sub-categories with one no name subcategory and one with prompt inversion?  It seems that prompt inversion does not belong to model inversion but should be put under Inversion Attacks, parallel to model inversion?
>
> The authors should avoid similar confusion.

---

> > ### Author Response · Authors · 2025-02-11
> > **Response to More Feedback**
> >
> > - We definitely wanted to convey that Transferable Attacks are a subset of Automated Attacks, and similarly, Prompt Inversion is a type of Model Inversion. Unfortunately, there is no clear way to represent that in TikZ.
> > - To avoid potential confusion, we have merged these boxes as per the reviewer's request.

---

> ### Comment · Reviewer_6QJR · 2025-02-11
> **Thanks for the revision for this week!**
>
> Hi authors,
>
> Thank you the authors for satisfying my constant revision harassment this week.
>
> One last question, for prompt inversion attack, doesn't it make more sense to parallel with model inversion attack? They should to be different in some senses.

---

> > ### Author Response · Authors · 2025-02-11
> > **Welcome the heated debate**
> >
> > - We welcome the reviewers' time engaging in this heated debate which has ended up improving the paper.
> > - For prompt inversion, prior literature has referred to prompt reconstruction as a kind of language model inversion. It can as well be placed in parallel with model inversion but we wanted to be consistent with prior literature. The rationale here is that no one uses a LM standalone in practical applications. It is always coupled with an application prompt. The model conditioned on the application prompt together form the model artifact and constitute the IP which LLM-based application providers would potentially want to protect. Here is the quote from [Language Model Inversion](https://arxiv.org/pdf/2311.13647) paper.
> > >We formalize this problem of prompt reconstruction as language model inversion, recovering the
> > input prompt conditioned on the language model’s next-token probabilities.

---

> ### Comment · Reviewer_6QJR · 2025-02-12
>
> After revision, I think all of my concerns have been carefully addressed. I recommend acceptance of this paper due to its comprehensiveness, covering almost all the attack vectors I know. The depth of the paper is also satisfactory. The authors convey their understanding to the overall attack settings and the technical side, and this apparantly cannot achieved without a thorough and large volumn of paper reviewing. The field is growing fast and we urgently need such a paper to be visible to researchers to accelerate safety research.  I believe this paper should have good value to the community.
>
> In addition, as the field is growing fast, I suggest the authors to continuously update the paper even after its acceptance. This might be useful for subsequent research.

---

### Review · Reviewer_Bwyk · 2025-01-16

**Summary Of Contributions:**

This paper presents a detailed SoK on existing attacks against large language models in a red-team setting. The author(s) present the attacks by introducing a taxonomy grouped by the difficulty level of achieving each attack’s entry point, starting from simple API access (jailbreaking) to more advanced gradient-based (given access to models … etc.). After discussing the notable papers in the literature on attacks, the authors then briefly summarize three categories of existing defense methodologies. Finally, the author(s) notes 6 key takeaways/findings from their survey work for red-teaming practitioners.

**Audience:**

Yes

**Broader Impact Concerns:**

none.

**Claims And Evidence:**

Yes

**Requested Changes:**

- can the authors flip the order of 6.1 and 7 —> first lead w/ challenges and future directions, as they should guide the takeaways that the authors want readers to leave with
    - by doing this, the takeaways in 6.1 can be adjusted to reflect the challenges and future directions, which adds more interesting / nuanced details to the takeaways
- can the authors address section 5 in more detail? namely how motifs in the existing defenses helps contextualize the key takeaways of red-teaming in section 6.1.
- is there anything the authors can add to the earlier sections that helps the reader differentiate between how LLM attacks are currently conducted/created vs. dedicated red-teaming efforts of LLMs? the current manuscript assumes they are the same but exploring this nuance might be of interest to readers.

**Strengths And Weaknesses:**

[strengths]

- comprehensive and detailed overview of existing attacks in the literature
- taxonomy is clear and well-adhered
- transition from the possible different attack surfaces (section 3) to the actual attacks (section 4) is clear and well-written

[weaknesses]

- the takeaways from 6.1 are complete and intuitive but can come across as more trivial than the authors likely desired
- takeaways and challenges/future directions are disjoint and talk about different things, though it makes a lot of sense for them to be related
- existing defenses section seems a bit out of place, as it is not talked about much in relation w/ the takeaways in 6.1
- lack of intuitive relationship between dedicated red-teaming, and more general efforts to attack LLMs

---

> ### Author Response · Authors · 2025-01-30
> **Response Reviewer 2**
>
> # Revisions Made in Response to Reviewer Comments
>
> ## 1. Reordering and Strengthening Takeaways
> - Reordered Sections 6.1 and 7 to lead with challenges/future directions before presenting takeaways
> - Completely rewrote takeaways section to ground recommendations in specific challenges and threats
> - Added concrete connections between takeaways and prior research
> - Enhanced practical value while maintaining systematic approach
>
> **Eg.**
> ```
> Domain-Specific Risk Taxonomy: While publicly available risk taxonomies provide solid founda- tions, practitioners should customize them based on domain-specific concerns (Section § 2.2). For example, hallucination risks are often under-emphasized in standard taxonomies but may be critical for domains like retrieval-augmented generation, legal analysis, and financial services where factual accuracy is paramount. Applications should adapt taxonomies to prioritize risks relevant to their use case while maintaining standard safety considerations.
>
> Diversity in Red-Teaming: Red-teaming exercises should combine both manual and automated approaches, as this enables greater semantic diversity in discovering vulnerabilities (Section § 6). This is particularly important given the increasing sophistication of attack prompts over time (Section § 4.1). Building and maintaining a community of trusted red-teamers can accelerate the identification of novel attack vectors (OpenAI, 2023).
>
> ```
>
> ## 2. Strengthening Defense-Takeaway Connections (Section 5)
> Added new content throughout Section 5 to better connect defense methodologies with red-teaming practices:
> - Added contextual paragraphs for each defense category
> - Explained how different defenses inform red-teaming practices
> - Demonstrated real-world applications of defense strategies
> - Included practical examples like:
>   - How perplexity filtering catches adversarial suffixes
>   - How blue teams iteratively strengthen defenses
>   - Role of guardrails in production environments
>
> **Eg.**
> ```
> From a red-teaming perspective, these prompt-based defenses represent the blue team’s first line of protection against Jailbreak Attacks (Section § 4.1) and Direct Attacks (Section § 4.2). These defenses span the full security capabilities matrix: prevention through input validation and filtering, detection via content moderation APIs, mitigation using fallback responses, and recovery through logging and incident response. Importantly, these extrinsic defenses require minimal access to the underlying model, making them particularly suitable for applications built on black-box LLM APIs. When red-teaming LLM applications, it is crucial to evaluate them with all guardrails activated, as this reflects real-world deployment conditions. Red teams can systematically probe defense effectiveness by testing different prompt attack strategies against multiple protective layers, while blue teams iteratively strengthen these defenses based on discovered vulnerabilities. This creates a continuous improvement cycle - for instance, while perplexity filtering may catch optimized adversarial suffixes, red teams might discover more sophisticated attacks using human-readable text that bypass this defense, prompting blue teams to implement additional protective measures like semantic analysis. Most current red-teaming research focuses on this iterative strengthening of defense mechanisms, particularly for these extrinsic defenses due to their broad applicability and ease of implementation in production systems.
> ```
>
> ## 3. Clarifying Red-Teaming vs General Attacks
> Added clarifying paragraphs throughout to differentiate systematic red-teaming from general attacks:
> - For Transferable Attacks: Explained value for systematic testing at scale
> - For Inversion Attacks: Detailed how privacy researchers can leverage techniques
> - For Infusion Attacks: Described evaluation across different risk types
> - For Backdoor Attacks: Outlined implementation challenges and competition insights
> - Emphasized structured methodology of red-teaming vs ad-hoc attack discovery
>
> These revisions strengthen the paper by:
> - Improving logical flow and impact
> - Creating clearer connections between sections
> - Providing more actionable insights
> - Better distinguishing between different security assessment approaches
>
> **Eg.**
> ```
> Such automated attack discovery methods are particularly valuable for red-teaming at scale, allowing systematic testing of model robustness across different threat vectors. While manual testing provides valuable insights, automated methods can efficiently explore large attack surfaces and identify vulnerabilities that might be missed in manual testing. In practice, security researchers with API access can leverage tools like AutoDAN or GCG algorithms, combined with proxy open-source models, to conduct these red-teaming exercises without requiring access to model internals.
> ```
>
> Detailed response: https://docs.google.com/document/d/1JYPmOqpTH_tMDHsh3mhIY4UY-K9QacbHKKC2nNJKBhY/

---

> > ### Comment · Reviewer_Bwyk · 2025-02-19
> > **Thank you for rebuttal**
> >
> > I want to thank the authors for the detailed rebuttal. The structure of the takeaways is much improved, and I appreciate the authors taking another stab at the takeaways section.
> >
> > All the changes have made the paper significantly stronger, and I have no other concerns.

---

### Review · Reviewer_6xzB · 2025-01-20

**Summary Of Contributions:**

The paper provides a systemization of knowledge (SoK) on red-teaming attacks against large language models. The presentation is done in a well-organized, hierarchical taxonomy classified based on the attacker access level (access to input, LLM API, in-context data, model inner activations, and training). The organization is reasonable and includes most of the significant existing literature on each type of attack. Then, there is also a devoted section on existing defense methods under three types (extrinsic, intrinsic, holistic).

Overall, the paper presents with a very comprehensive analysis which covers recent works in red-teaming attacks against LLMs, providing a great reference for the community.

**Audience:**

Yes

**Claims And Evidence:**

Yes

**Requested Changes:**

- Section 5 could be written in a way that is more connected with Section 4, namely the hierarchical taxonomy.
- Section 6.1 could be connected with other sections more smoothly. Maybe explain each takeaway with slightly more detail than simply referencing a related section number?
- The term "training-time attack" or "gradient-based attack" could be rephrased for clarity. Maybe change the hierarchy a little bit? Access to model weight and access to model training seem a bit disjoint; it could instead be a two-level hierarchy such as "White-box attack (full access to model weights)" and "training-time attack" could be its subcategory, because it is a stronger access assumption. Then again, backdoors are not white-box attacks so perhaps this category should be split into two: (1) training attack, and (2) white-box attack (although this is still not very clear-cut).

**Strengths And Weaknesses:**

**Strengths:**
- Well-organized taxonomy based on the adversarial access level provides an intuitive hierarchical structure.
- Very comprehensive and exhaustive coverage of the literature, including most (if not all) existing works.
- Figure 1 and 2 are both very easy to understand and comprehensive.
- The paper is overall very well written (esp. Section 1 and 4), which is essential for an SoK work.

**Weaknesses:**
- Although the paper positions as a red-teaming SoK, a significant part of the work covers targeted attacks which are not necessarily a red-teaming attack. Although they can be used as a red-teaming technique, the individual works may not be focused on red-teaming. It would be better if this work, as a red-teaming SoK, discussed how they can be used for red-teaming or explained the differences.
- Defense section feels slightly less connected to the attacks section, as it simply lists the current literature in defense against the discussed attack methods. A structure that is more aligned with the attacks section could strengthen the connection between sections.
- Gradient-based attack is categorized under Training-time attack, which may come across as confusing. The terminology suggests that the attack is done during training time but Gradient-based attack only requires white-box access to the model, which does not necessarily mean a training-time access.

---

> ### Author Response · Authors · 2025-01-30
> **Response Reviewer 3**
>
> # Response to Review
>
> We sincerely appreciate the thoughtful and constructive feedback. Please find our response below.
>
> ### On Attack Methods and Red-Teaming
> We've thoroughly revised each attack section by adding paragraphs with each subsection to explicitly show how security teams can use these techniques in red-teaming exercises. For instance, we now explain how manual jailbreak studies help identify gaps in content filtering, and how automated methods enable systematic testing at scale. We've added practical guidance throughout - from using privacy research techniques to assess vulnerabilities, to testing retrieval systems, to evaluating compound LLM architectures.
>
> ### Strengthening the Attack-Defense Connection
> While we considered directly mirroring the attack taxonomy, we realized this wouldn't capture how most defenses protect against multiple attack vectors simultaneously. Instead, we've rebuilt this section to show these relationships more clearly, with specific examples of how each defense applies to different attacks. We've also added practical implementation guidance and examples of how defenses evolve through iterative red-teaming.
>
> ### Attack Categorization
> Thank you for catching the potential confusion in our categorization. We've made the taxonomy clearer by:
> - Renaming categories to focus on access levels rather than timing
> - Splitting training attacks into "backdoor" (data poisoning) and "frontdoor" (direct model manipulation)
> - Clarifying that white-box prompt search is distinct from training-time attacks
>
> ### Takeaways
> We've completely rewritten this section to provide concrete, actionable guidance rather than just section references. The new takeaways cover crucial aspects like adapting risk taxonomies for specific domains, considering the full attack surface beyond just prompt-based attacks, and implementing proper evaluation protocols. Each recommendation now includes specific examples and practical implementation steps.
>
> These revisions make the paper more immediately useful for practitioners while maintaining its theoretical rigor. Thank you for helping us improve the work's clarity and practical value.
>
> Please see examples for these changes in response to reviewer 2 and the full list of additional paragraphs in the following document
>
> **Detailed Response** https://docs.google.com/document/d/1QDUpghGIu-jTJLkCzaWd9Bh7fFFJrOPbcrooOl1ss88/

---

> > ### Comment · Reviewer_6xzB · 2025-02-06
> >
> > Thank you for the detailed rebuttal. All of my previous concerns have been addressed.
> >
> > I think the new taxonomy (the Training Attack part) feels a lot more natural and reasonable. Categorizing it into two groups, namely Backdoor attack and Frontdoor attack, makes a lot of sense to me. The new takeaway section is also more comprehensive and self-contained compared to the previous version.
> >
> > Overall, I think the paper is much stronger after the revision. I commend the effort the authors have put into making revisions of the paper and addressing the concerns.

---

### Author Response · Authors · 2025-01-30
**Summary of Changes in Response to Reviews**

We thank the reviewers for their thorough, constructive, and insightful feedback that has significantly improved our paper. Based on their comments, we identified and addressed four key themes:

1. Better articulating the relationship between general LLM attacks and dedicated red-teaming efforts
2. Strengthening connections between defense methodologies and specific attack vectors
3. Adding more depth and nuance to our key takeaways
4. Clarifying the organization and terminology around gradient-based and training attacks

Our revisions include:
* Added contextualizing paragraphs throughout the attack sections to better connect them with red-teaming practices
* Introduced new content linking each defense strategy to specific attack vectors and red-teaming applications
* Completely revised the takeaways section to provide deeper insights and more nuanced recommendations
* Reorganized the training attacks section to more clearly distinguish between data-centric (Backdoor) and model-centric (Frontdoor) approaches
* Added detailed tables explicitly describing adversary goals and capabilities in the threat model

Given the rapid evolution of this field, we have incorporated many of the excellent paper suggestions provided by Reviewer 1, along with other significant works published since our initial submission. We appreciate the reviewers helping us stay current with the latest developments in this fast-moving area. We have already incorporated several papers from Reviewer 1's list and will add more to the camera-ready version.

We believe these revisions have substantially strengthened the paper while maintaining its core contribution of providing a systematic framework for understanding and implementing LLM security measures.

**Additional Improvements:**

We have added a new section highlighting the crucial distinction between safety and security in LLMs:

*Safety vs Security*

Safety and security represent distinct objectives in LLM risk management (Qi et al., 2024a). Safety focuses on preventing harm that LLMs might inflict upon their environment - for example, generating toxic content or providing harmful advice accidentally. Security focuses on protecting LLMs against malicious exploitation - such as preventing jailbreaking attacks or unauthorized access to training data.

These objectives entail different threat models:
* Safety primarily addresses non-adversarial scenarios like unintended model behaviors and inherent flaws
* Security explicitly considers adversarial scenarios where malicious actors attempt to compromise the system

Our threat model systematically analyzes attack surfaces and entry points that adversaries may exploit, from jailbreaking attempts at the application layer to more sophisticated attacks targeting model weights and training processes. This security-oriented approach complements existing safety research while specifically focusing on defending against malicious exploitation.

To **facilitate easy review of these changes** we also provide a **diff of the old and the new pdfs** which can be accessed at: https://draftable.com/compare/rzxkvUeosujU

We have also updated the pdf with the new pdf in the original submission.

---

### Author Response · Authors · 2025-03-04
**Decision Request**

Dear Action Editor,

I'm writing to kindly follow up on our paper submission. We recently inquired about its status through OpenReview but haven't received a response.

Could you please inform us of the decision on our submission?

Thank you for your time.

---

### Decision · Action_Editor_cE9K · 2025-03-27

**Recommendation:** Accept as is

**Comment:**

This paper presents a comprehensive SoK on red-teaming attacks against large language models. Initially the reviewers raised concerns about the structure and the clarity of the paper, particularly regarding the taxonomy of attacks (6QJR, 6xzB). However, the authors engaged in a thorough revision process, and directly addressed these concerns, including a significant restructuring of the taxonomy, adding new sections and paragraphs about connecting the attack types to red-teaming practices and linking the defense methods to attack vectors, especially interacting with Reviewer 6QJR.

Following the substantial revision, all reviewers reported that their concerns have been addressed, agreeing on the acceptance of the paper. The revised paper provides a well-supported systemization of the knowledge in this area.

**Audience:**

Yes

**Claims And Evidence:**

Yes